# Predictors of homebirth amidst COVID-19 pandemic among women attending health facilities in Wondo Genet, Sidama Region, Ethiopia: A case control study

**Asaminew Geremu Gaga, Teshome Abuka Abebo, Yilkal Simachew** *

School of Public Health, College of Medicine and Health Science, Hawassa University, Hawassa, Ethiopia

* joemakalister123@gmail.com

## Abstract

### Background

In developing countries, home delivery increases the risk of maternal and perinatal mortality. Despite this, home deliveries account for a considerable share of deliveries in developing nations such as Ethiopia. Evidence on factors that affect homebirth is required for the measures needed to overcome these conditions.

### Objective

To identify predictors of homebirth among women attending health facilities in Wondo Genet, Sidama Region.

### Methods

Unmatched case-control study was conducted from May to June 2021 among 308 mothers (102 cases and 206 controls) who recently delivered and visited either postnatal care or sought immunization service at public health facilities of Wondo Genet. A structured interviewer-administered questionnaire was used to collect data. Epi-Data version 3.1 was used for data entry, and the Statistical Package for the Social Sciences (SPSS) version 20 was used for data analysis. Bivariate and multivariate logistic regression analyses were used to identify the determinants of homebirth. The association between the outcome variable and independent variables was declared statistically significant at a P-value < 0.05 with a 95% Confidence Interval (CI) in a multivariable model.

### Results

Rural residence [AOR: 3.41; 95%CI: 1.58–7.39], lifetime physical IPV [AOR: 2.35; 95%CI: 1.06–5.17], grand-multiparity [AOR: 5.36; 95%CI: 1.68–17.08], non-use of contraception before recent pregnancy [AOR: 5.82; 95%CI: 2.49–13.60], >30 min to reach health facility [AOR: 2.14; 95%CI: 1.02–4.51], and lack of facemask [AOR: 2.69; 95%CI: 1.25–5.77] were statistically significant predictors of homebirth.

**Data Availability Statement:** All relevant data are within the paper and its Supporting Information files.

**Funding:** The author(s) received no specific funding for this work.

**Competing interests:** The authors have declared that no competing interests exist.

## Conclusion and recommendation

The access gap to maternity services should be narrowed between rural and urban women. Healthcare programs concerning women's empowerment could help reduce persistent intimate partner violence. Family planning needs to be promoted, and multiparous women should be counseled on the adverse obstetric consequences of homebirth. The devastating effect of the coronavirus disease 2019 pandemic on maternity services should be prevented.

## Introduction

Homebirth refers to childbirth practice outside a healthcare facility without necessary setup [1]. Unskilled birth at home can threaten the life of both mother and baby due to pregnancy and childbirth-related complications that are unpredictable but preventable with access to quality skilled care [2, 3]. Most maternal and perinatal mortality is related to a lack of skilled care during and immediately after childbirth [3–5].

According to World Health Organization (WHO), about 810 women died every day of pregnancy and childbirth-related causes in 2017 [6]. Around 86% of these deaths occurred in developing countries, particularly in Sub-Saharan Africa and South Asia. Globally, about three-quarters of all maternal deaths are related to life-threatening complications, including hemorrhage, hypertensive disorders of pregnancy, and infection that can occur during pregnancy, delivery, and the postpartum period [3, 4, 7]. Traditional Birth Attendant (TBAs) can neither predict nor have the necessary skill, equipment, and medicine to cope with these complications during homebirth [2]. Apart from increasing maternal mortality, home delivery is associated with the highest perinatal deaths and maternal morbidity, like obstetric fistula [5, 8].

The reduction of maternal mortality was given due attention when it became one of the eight health-related Millennium Development Goals (MDGs) targets in 2000; notably, MDG 5 was set to reduce maternal mortality by 75% from 1990 to 2015 [9]. However, there has yet to be much progress in many developing countries including Ethiopia, and the strategic transition to Sustainable Development Goals (SDGs) has taken place without achieving MDGs targets by 2015 with high maternal mortality [3]. In most Sub-Saharan African countries, the highest proportion of homebirths had been a significant impediment to achieving MDG 5 [10].

Globally, about 71% of births occur in the presence of skilled attendants, and about one-third of births occur at home [11]. Unskilled home delivery is more common in resource-poor settings, with the highest maternal mortality [2, 6, 10]. In Sub-Saharan Africa, about 34% of women give birth at home, with remarkable disparities across the countries; hence, Chad, Ethiopia, Niger, and Nigeria recorded the highest proportion of homebirth with 76.8%, 68.2%, 66.6%, and 59.1%, respectively [12]. According to Ethiopian Mini Demographic and Health Survey (EMDHS) in 2019, about 52% of women give birth at home, with marked variation across regions [13]. Several small studies in different regions of Ethiopia have also reported a high prevalence of home delivery, ranging from 21.2% in Bahirdar city, Northwest Ethiopia, to as high as 80% in Sherkole District, West Ethiopia [14–17].

Most rural women in Sidama Region (formerly known as Sidama Zone) face difficulty in accessing distant health facilities to use maternal health services [18, 19]. In addition, poor landscape and topography make institutional delivery less reliable in this region. In previous studies, several socio-demographic and economic factors, including age, parity, place of

residence, lack of education, poor wealth index, religion, media exposure, and autonomy, were associated with home delivery [20–24]. Antenatal care (ANC) attendance, the intention of pregnancy, birth preparation, knowledge of obstetric danger signs, perceived quality of health services, and the time taken to reach the health facilities were also determinants of home delivery identified by other studies [22, 24–27].

The Ministry of Health has been encouraging all pregnant women to have skilled delivery by expanding healthcare facilities with emergency obstetric care services and providing all maternity services free of charge [28]. As a result, there has been remarkable progress in the proportion of institution-based deliveries over time, from 5% in 2005 to 48% in 2019 [13]. Nevertheless, home delivery is still common, with more than half of births at home. In Wondo Genet, the study area, the prevalence of homebirth was 38.8% in 2017 [29]. However, little is known about Intimate Partner Violence (IPV) and barriers associated with the Coronavirus Disease 2019 (COVID-19) pandemic as independent predictors of homebirth. This study aimed to identify predictors of homebirth among women attending public health facilities in Wondo Genet, Sidama Region, Ethiopia, 2021. The results of this inquiry will enable the local health management teams, policymakers, and other relevant bodies to plan new strategies to come up with solutions to many barriers that hinder maternal delivery services utilization.

## Methods and materials

### Study area

The study was conducted in Wondo Genet, approximately 270 kilometers south of Addis Abeba and 24 kilometers east of Hawassa City. Wondo Genet is one of the tourism destinations in the Sidama region and located 1880 meters above sea level. Wondo Genet has five health centers: Babo-Chorora, Wosha-Soyama, and Aruma are administered by the District (rural) government, while Chuko and Wondo Genet 01 are administered by the Town. Despite there are some private clinics (medium and primary) at wondo Genet, none of them provide delivery service. According to a 2018 estimate, the total population of Wondo Genet was 423,244, of whom 210,781 are female, and 212,463 are male.

### Study design and population

Unmatched case-control study was conducted from May 19 to June 11, 2021, among women attending public health facilities in Wondo Genet. The source population was all women who had given birth in the last 12 months before data collection. While the study population was all selected women who had given birth in the last 12 months before data collection and who visited selected public health facilities for child immunization and/or Postnatal care (PNC) in Wondo Genet. Cases were women who had delivered at home. Controls were women who had delivered in the healthcare facilities.

### Inclusion and exclusion criteria

Women who delivered at home or residence (cases) and in a health facilities (controls) in the last 12 months and visited public health facilities for child immunization and/or PNC in Wondo Genet were included. The study excluded women who delivered on the way to the health facility.

### Sample size and sampling procedures

The sample size was calculated using Stat Calc of EPI Info Version 7 Statistical Software using a two-proportion formula. The following assumptions were used to estimate the sample size

needed for the study. The exposure variable was the time required to reach the health facility [30], and the proportion of cases and controls exposed was 28.2% and 13.0%, respectively. In addition, after considering 80% power, 95% confidence level, 1:2 case-to-control ratio, and 15% non-response rate, the final sample size was 308 (102 cases and 206 controls). Three of the five health centers were selected by simple random sampling. After estimating the average number of mothers coming to each health center for child immunization and/or postnatal care services from the previous reports, the sample was allocated proportionally to each health facility. Mothers who gave birth within the previous twelve months were asked about their delivery place and classified as cases or controls. Consecutive sampling was employed to select study participants. Hence, two consecutive controls were selected after each case was selected until the predetermined sample was attained from each health facility. The survival sampling/ cumulative density sampling method was used in this study to select the control.

## Study variables

**Dependent variable.** Home delivery was the dependent variable (the dependent variable was dichotomized into home birth = 1 and facility birth = 0).

**Independent variables.** *Socio-demographic characteristics*. Age, parity/birth order, place of residence, education, occupation, income, media exposure, ethnicity, religion, marital status, women's autonomy, and experience of IPV.

*Health facility and providers related factors*. Time taken to reach health facility, perceived quality of services, and gender preferences for a birth attendant.

*Obstetrics and maternal healthcare-related factors*. Family planning, the intention of pregnancy, Antenatal Care (ANC), experience of obstetric complications, birth preparation, and knowledge of obstetric danger signs.

*COVID-19 related factors*. Fear of exposure to the disease, lack of transport, lockdown, and lack of face mask.

The women's knowledge of obstetric danger signs was measured using five questions for danger signs during the postpartum period, with a minimum score of 0 and a maximum score of 5. Six questions were asked to measure women's knowledge of danger signs during pregnancy, childbirth, and the neonatal period with a minimum score of 0 and a maximum score of 6. Finally, women who spontaneously mentioned $\geq$75%, 50–74.9%, and less than 50% of danger signs at each obstetric phase were considered to have good, fair, and poor knowledge, respectively.

Perceived quality of health services was measured in terms of individual perception of waiting time, protection of privacy, and the level of health providers' interaction with clients. A score of '1' was given for each factor favoring good quality, such as short waiting time (in this case, <30 minutes), protect privacy, and smooth patient-provider interactions. On the other hand, long waiting time ($\geq$30 minutes), unprotected privacy, and poor patient-provider interactions were given a score of '0'. Women who scored '2–3' and '$\leq$1' were said to have a perception of good and poor quality, respectively.

## Data collection procedure and tool

Data were collected using a structured interviewer-administered questionnaire adapted from other related studies. The questionnaires include socio-demographics, health facilities and providers, obstetrics and maternal healthcare, COVID-19 related factors, and reasons for a home birth. Data collection was undertaken by three diploma nurses and supervised by two health officers.

## Data processing and analysis

Data were entered and cleaned using Epidata Version 3.1, then cleaned and exported to Statistical Package for Social Sciences (SPSS) Version 20 (SPSS Inc., Chicago Illinois, USA) for analysis. Data were summarized by descriptive statistics and presented in texts, tables, and graphs.

Binary logistic regression model was run to assess the association between dependent and independent variables. A bivariate analysis was done to select candidate predictor variables for a multivariable model at p-value <0.25. The confounders were controlled for by running a multivariable logistic regression model. A tolerance <0.4 and Variance Inflation Factor (VIF) >2.5 were used as cut-off values to determine multicollinearity. Further, a correlation matrix was also used to identify specific variables that were correlated with each other. The two-way interactions among included factors were tested in the adjusted model, and there were no significant interactions. The model was deemed fit at P-value >0.05 on the Hosmer and Lemeshow goodness of fit test. The association between the outcome variable and independent variables was declared statistically significant at P-value <0.05 with a 95% Confidence Interval (CI) around Adjusted Odds Ratio (AOR) in a multivariable model.

## Data quality control

The questionnaire was developed in English but later translated into Sidaamu Afoo by a language expert. Then, it was back-translated into English to assure its clarity and consistency. The principal investigator trained data collectors and supervisors on study-related issues. Pretest was conducted on 5% (n = 16) of the sample in Gara-Rikata Health Center (the neighboring health center in Tula Sub-city of Hawassa) one week before the actual data collection.

## Ethical consideration

This study was conducted after obtaining ethical clearance from the Institutional Review Board of Hawassa University, College of Medicine and Health Sciences (Reference No: IRB/156/13). Considering the non-invasive nature of the data collection procedure and the literacy status of the study area, verbal consent which IRB approved, was obtained from all study subjects after a detailed explanation of the study objectives and the right to withdraw from the study at any time. The data collector read the information sheet and consent to the participants slowly and loudly. Then they were asked if they had any questions. After the mothers confirmed everything was clear, they were asked one last time if they wanted to participate in the study. The data collectors circled either 'yes' or 'no' depending on which option was chosen. The interview was only conducted if the data collector was instructed to circle the response 'yes'.

Nine (2.9%) mothers (3 cases and 6 controls) were under 18 years old, but consent was obtained from them because they were married and did not live with the family. The IRB was informed of this circumstance and gave its approval for the verbal consent received from mothers who were under the age of 18. In addition, confidentiality was assured by excluding their name during the data collection period, and the participant's data were handled with strict anonymity.

## Results

### Socio-demographic characteristics of respondents

Among the 308 women selected, all respondents (102 cases and 206 controls) were successfully interviewed with a response rate of 100%. About 236 (76.6%) participants were between the age range of 15 to 29 years, with a mean age (± standard deviation) of 26.80 (±5.04) years and

25.06 (±4.76) years for cases and controls, respectively. Over three-fourths, (78.4%) of cases and nearly one-third (32%) of controls were rural residents. More than half (52.0%) of cases were uneducated, and almost a similar percentage (50%) of controls had primary education. By parity, many cases (55.9%) were grandmultiparous, and a significant proportion of controls (59.7%) were low multiparous. Eighty-one (79.4%) cases and 73 (35.4%) controls had ever experienced physical IPV (**Table 1**).

## Health facilities, providers, and COVID-19 related factors

Seventy-one (69.6%) cases and 68 (33.0%) controls were reported to reach the nearest health facility taking at least thirty minutes. More than half (57.8%) of cases and three-quarters (75.2%) of controls rated the quality of care they received in a health facility as good. Regarding COVID-19 related factors, it was noted that 49 (48.0%) cases and 91 (44.2%) controls were scared about contracting COVID-19 if they go to a health facility. During the COVID-19 pandemic, thirty-two (31.4%) cases and 28 (13.6%) controls had experienced difficulty in using health services due to lack of transport when pregnant with the last child. Similarly, thirteen (12.7%) cases and 19 (9.2%) controls had faced difficulty in accessing health services during pregnancy due to the lockdown. Also, more than two third (67.6%) of cases and about a quarter (25.2%) of controls had faced difficulty in accessing health services due to lack or non-use of facemasks (**Table 2**).

## Obstetrics and maternal healthcare-related factors

Twenty-three (22.5%) cases and 124 (60.2%) controls had used any form of modern contraception before the most recent pregnancy. Sixty-nine (67.6%) cases and 151 (73.3%) controls had intended pregnancy. A significant proportion of cases (78.4%) and controls (69.9%) had not experienced obstetric complications during the most recent childbirth. Among those who attended ANC, more than one-half (57.4%) of cases and over two-fifths (43.0%) of controls initiated their first ANC visit in second trimester of pregnancy. Therefore, 34 cases and 27 controls had missing data on gestational age at first ANC attendance for women who did not visit ANC. The majority of cases (62.7%) and controls (58.7%) did not prepare before the last birth. Concerning women's knowledge of obstetric danger signs, plenty of cases and controls had poor knowledge of pregnancy, labour and delivery, postpartum, and neonatal danger signs (**Table 3**).

## Reasons for homebirth among cases

From 102 cases, nearly half (48%) of respondents justified their reason for home delivery as "sudden onset of labour". Fourteen respondents (13.73%) mentioned that childbirth in a health facility is unnecessary unless they face difficulty during labour and delivery. Six women (5.88%) justified that "homebirth is my usual practice" (**Fig 1**).

## Predictors of homebirth amidst the COVID-19 pandemic

As shown in Table 4 below, rural residence, grand multi-parity, ever experience of physical IPV, >30 min to reach the nearest health facility, non-use of family planning before the most recent pregnancy, and lack of facemask during the COVID-19 pandemic were significant predictors of homebirth. Rural women were about 3.4 times more likely to deliver at home than their urban counterparts [AOR: 3.41; 95%CI: 1.58–7.39]. Women who are grand-multiparous had about 5.4 times increased likelihood of delivering at home than those who are primiparous [AOR: 5.36; 95%CI: 1.68–17.08]. Ever experience of physical IPV increased the odds of

**Table 1. Socio-demographic characteristics of respondents in Wondo Genet, Sidama Region, Ethiopia, 2021 (N = 308).**

| Variables | Category | Cases | Controls |
|---|---|---|---|
| | | Frequency (%) | Frequency (%) |
| Maternal age | Mean (±SD) years | 26.80 (±5.04) | 25.06 (±4.76) |
| | 15–24 Years | 35 (34.3) | 87 (42.2) |
| | 25–29 Years | 44 (43.1) | 70 (34.0) |
| | ≥30 Years | 23 (22.6) | 49 (23.8) |
| Place of residence | Urban | 22 (21.6) | 140 (66.0) |
| | Rural | 80 (78.4) | 66 (32.0) |
| Maternal education | Uneducated | 53 (52.0) | 19 (9.2) |
| | Primary education | 38 (37.3) | 103 (50.0) |
| | Secondary and above | 11 (10.7) | 84 (40.8) |
| Maternal occupation | Housewife | 65 (63.7) | 94 (45.6) |
| | Merchant | 17 (16.7) | 55 (26.7) |
| | Employer | 5 (4.9) | 26 (12.6) |
| | Others[1] | 15 (14.7) | 31 (15.0) |
| Monthly income | <500 ETB | 27 (26.5) | 62 (30.1) |
| | 500–1499 ETB | 38 (37.3) | 62 (30.1) |
| | ≥1500 ETB | 37 (36.2) | 82 (39.8) |
| Husbands education (N = 299) | Uneducated | 31 (30.7) | 63 (31.8) |
| | Primary education | 44 (43.6) | 74 (37.4) |
| | Secondary and above | 26 (25.7) | 61 (30.8) |
| Husbands occupation(N = 299) | Farmer | 53 (52.5) | 79 (39.9) |
| | Merchant | 32 (31.6) | 72 (36.4) |
| | Employer | 4 (4.0) | 21 (10.6) |
| | Others[2] | 12 (11.9) | 26 (13.1) |
| Religion | Protestant | 62 (60.8) | 112 (54.4) |
| | Islam | 23 (22.5) | 44 (21.4) |
| | Orthodox | 5 (4.9) | 38 (18.4) |
| | Others[3] | 12 (11.8) | 12 (5.8) |
| Media exposure | Never | 39 (38.2) | 65 (31.6) |
| | Sometimes | 40 (39.3) | 87 (42.2) |
| | Always | 23 (22.6) | 54 (26.2) |
| Autonomy on a place of delivery | Mother | 26 (25.5) | 43 (20.8) |
| | Joint | 42 (41.2) | 99 (48.1) |
| | Husbands/others | 34 (33.3) | 64 (31.1) |
| Physical IPV during pregnancy | Yes | 34 (33.3) | 58 (28.2) |
| | No | 68 (66.7) | 148 (71.8) |
| Sexual IPV during pregnancy | Yes | 32 (31.4) | 53 (25.7) |
| | No | 70 (68.6) | 153 (74.3) |
| Emotional IPV during pregnancy | Yes | 61 (59.8) | 70 (34.0) |
| | No | 41 (40.2) | 136 (66.0) |

ETB: Ethiopian Birr, IPV: Intimate Partner Violence, Others: Daily labourer, Maidservant

Others[2]: Daily labourer, Driver, Barber, Garage worker, Others[3]: Catholic, Adventist and Traditional

homebirth by 2.3 fold [AOR: 2.35; 95%CI: 1.06–5.17]. Similarly, women who needed >30 minutes to reach the nearest health facility were about two times more likely to deliver at home than those who needed the utmost 30 minutes [AOR: 2.14; 95%CI: 1.02–4.51]. Women who

**Table 2. Health facility, providers, and COVID-19 related factors among cases and controls in Wondo Genet, Sidama Region, Ethiopia, 2021 (N = 308).**

| Variables | Category | Cases | Controls |
|---|---|---|---|
| | | Frequency (%) | Frequency (%) |
| Time taken to health facility | <30 min | 31 (30.4) | 138 (67.0) |
| | ≥ 30 min | 71 (69.6) | 68 (33.0) |
| Perceived quality of services | Good | 59 (57.8) | 155 (75.2) |
| | Poor | 43 (42.2) | 51 (24.8) |
| Gender preferences for birth attendants | Female | 47 (46.1) | 72 (35.0) |
| | Male | 15 (14.7) | 46 (22.3) |
| | Indifferent | 40 (39.2) | 88 (42.7) |
| Feared COVID-19 | Yes | 49 (48.0) | 91 (44.2) |
| | No | 53 (52.0) | 115 (55.8) |
| Lack of transport due to COVID-19 | Yes | 32 (31.4) | 28 (13.6) |
| | No | 70 (68.6) | 178 (86.4) |
| Lockdown | Yes | 13 (12.7) | 19 (9.2) |
| | No | 89 (87.3) | 187 (90.8) |
| Lack of face mask | Yes | 69 (67.6) | 52 (25.2) |
| | No | 33 (32.4) | 154 (74.8) |

COVID-19: Coronavirus Disease 2019

did not use contraceptive methods were about 5.8 times more likely to deliver at home than those who used contraceptive methods before the most recent pregnancy [AOR: 5.82; 95%CI: 2.49–13.60]. Lack of facemasks increased the odds of homebirth by nearly 2.7 fold [AOR: 2.69; 95%CI: 1.25–5.77].

## Discussion

Literature reveals that several factors have been cited to affect homebirth. However, it is important to recognize specific factors relevant in different settings as these may vary substantially. These factors should be emphasized in the design of interventions to reduce the proportion of homebirths and increase deliveries taking place at healthcare facilities.

In this study, parity was a significant predictor of home delivery. In particular, grand-multiparous women were about 5.4 times more likely to deliver at home than primiparous women [AOR: 5.36; 95%CI: 1.68–17.08]. The result aligns with the findings of other studies conducted in Nigeria, Ethiopia, Eritrea, and Nepal [20, 31–33]. A study done in Malawi demonstrated insignificant association of parity with homebirth [22]. This might be due to disparities in the study settings and the age composition of women, as the study included predominantly younger (15–24 years) and middle (25–34 years) reproductive age groups. Study conducted in Ethiopia also found contradicting finding [16]. It was argued that women with lower parity might anticipate and fear possible difficulties during labour due to a lack of experience at childbirth [33]. On the other hand, multiparous women might consider childbirth as their usual practice and think of being multiparous as though they would have less risky labour and delivery [31]. Multiparous women are usually older, and are more likely to be persuaded by stereotyped traditions from their descendants that facility delivery is unnecessary and that the services of TBAs are better than modern health services.

Regarding exposure to IPV, the women's experience of physical IPV retained its statistically significant association with homebirth. More specifically, women who experienced physical IPV in their lifetime had about 2.4 times increased odds of having homebirth than their

**Table 3. Obstetrics and maternal health care related factors among cases and controls in Wondo Genet, Sidama Region, Ethiopia, 2021 (N = 308).**

| Variables | Category | Cases | Controls |
|---|---|---|---|
| | | Frequency (%) | Frequency (%) |
| Family planning | Yes | 23 (22.5) | 124 (60.2) |
| | No | 79 (77.5) | 82 (39.8) |
| Intention of pregnancy | Intended | 69 (67.6) | 151 (73.3) |
| | Unintended | 33 (32.4) | 55 (26.7) |
| Experience of obstetric complications | Yes | 22 (21.6) | 62 (30.1) |
| | No | 80 (78.4) | 144 (69.9) |
| Frequency of ANC | <4 visit(s) | 79 (77.5) | 148 (71.8) |
| | ≥4 visits | 23 (22.5) | 58 (28.2) |
| GA at first ANC visit (N = 247) | 1st Trimester | 11 (16.1) | 93 (52.0) |
| | 2nd Trimester | 39 (57.4) | 77 (43.0) |
| | 3rd Trimester | 18 (26.5) | 9 (5.0) |
| Birth preparation | Prepared | 38 (37.3) | 85 (41.3) |
| | Unprepared | 64 (62.7) | 121 (58.7) |
| Knowledge of pregnancy danger signs | Poor | 73 (71.5) | 80 (38.8) |
| | Fair | 16 (15.7) | 55 (26.7) |
| | Good | 13 (12.7) | 71 (34.5) |
| Knowledge of labour and delivery danger signs | Poor | 52 (51.0) | 98 (47.6) |
| | Fair | 24 (23.5) | 50 (24.2) |
| | Good | 26 (25.5) | 58 (28.2) |
| Knowledge of postpartum danger signs | Poor | 69 (67.6) | 130 (63.1) |
| | Fair | 23 (22.5) | 62 (30.1) |
| | Good | 10 (9.8) | 14 (6.8) |
| Knowledge of neonatal danger signs | Poor | 71 (69.6) | 89 (43.2) |
| | Fair | 11 (10.8) | 48 (23.3) |
| | Good | 20 (19.6) | 69 (35.5) |

ANC: Antenatal Care

GA: Gestational Age

counterparts [AOR: 2.35; 95%CI: 1.06–5.17]. This result is consistent with a systematic review and meta-analysis conducted by Musa A. et al. (2019) [34]. In a secondary analysis of the Demography and Health Survey of low and middle-income countries and propensity score matching analysis in Nigeria, exposure to IPV reduced the proportion of deliveries assisted by skilled health professionals in a health facility [35, 36]. Experience of IPV can affect the women's psychosocial situation and health-seeking behavior that encourages the use of maternal health services [36]. Moreover, the women's experience of violence might limit decision-making power and restrict their freedom of movement for social interactions as well as utilization of maternal health services unless it is per their partner's permission. In this study, however, most of the variables related to IPV hadn't shown a statistically significant association with home delivery. This could be because of two reasons: firstly, most women popularly believe that men have the right to beat their wives, particularly when they become reluctant to do what their husbands want [37]. Secondly, women could be suspicious about disclosing such violence because they might consider it as if they betrayed their husbands in case certain legal concern arises.

Non-use of family planning showed a positive association with homebirth. In this study, women who did not use family planning before the most recent pregnancy had about 5.8

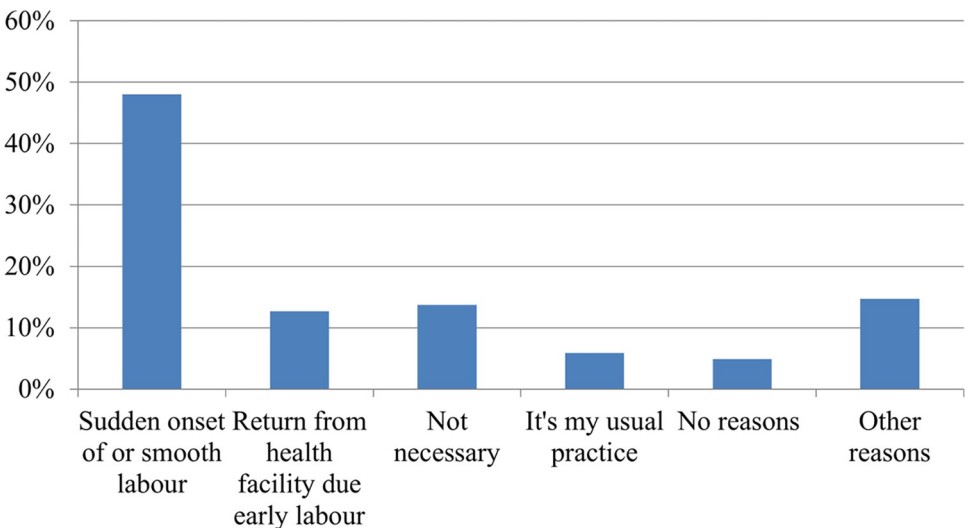

**Fig 1. Diagrammatic representation of reasons for homebirth among cases in Wondo Genet, Sidama Region, Ethiopia, 2021.**

times increased likelihood of delivering at home than women who did use any contraceptive methods [AOR: 5.82; 95% CI: 2.49–13.60]. The result coincides with the findings of studies conducted in Malawi and Ethiopia [22, 38]. Women who use contraceptive methods might have an increased tendency to attend ANC clinics and implement the advice received from health providers to have healthy obstetric outcomes, including using health facilities for child-birth as a continuum of care [22]. It could also be explained in such a way that women who use contraceptive methods might be familiar with health workers and other maternal health ser-vices, including skilled delivery; as a result, they adopt better health-seeking behavior and feel confident during the intimate procedures. Furthermore, the use of modern methods of contra-ception could be of prime importance in reducing maternal mortality. According to WHO, better access to modern contraceptive methods and quality obstetric care before and after childbirth can reduce the risk of maternal death [3]. Evidence has shown that contraceptive methods prevent unwanted pregnancies with an associated reduction of unsafe abortion, which is among the major causes of maternal mortality worldwide [39]. In view of this study, family planning might help reduce maternal death by decreasing the proportion of homebirths.

The place of residence showed a statistically significant association with homebirth; after accounting for other covariates in multivariable logistic regression, rural women were 3.4 times more likely to deliver at home compared to the women residing in urban areas [AOR: 3.41; 95%CI: 1.58–7.39]. The result was in line with several studies conducted in developing countries, including Ethiopia [12, 20, 21, 40]. However, some studies; for example, a case-con-trol study in the Gambela region, Southwest Ethiopia, and a prospective cohort study in Wolayta Zone, South Ethiopia, found a non-significant association between homebirth and place of residence [24, 41]. The variations in the study findings could be attributed to the dif-ferences in the study settings. Access to rural healthcare services and facilities is usually limited due to poor socio-economic status and topographical challenges [20]. This rural-urban dispar-ity could also be related to inequities in the distribution of healthcare infrastructure; indeed, rural residence in most developing countries is associated with more limited distribution of healthcare facilities compared to urban residence [22]. The physical location of healthcare

**Table 4. Bivariate and multivariate analyses of predictors of homebirth in Wondo Genet, Sidama Region, Ethiopia, 2021 (N = 308).**

| Variables | Category | Cases | Controls | COR (95%CI) | AOR (95%CI) | P-value |
|---|---|---|---|---|---|---|
| Women's education | Uneducated | 53 | 19 | 21.30 (9.39–48.28) | 2.16 (0.67–7.04) | 0.199 |
| | Primary education | 38 | 103 | 2.82 (1.36–5.85) | 1.52 (0.57–4.00) | 0.402 |
| | Secondary and above | 11 | 84 | 1 | 1 | |
| Place of residence | Urban | 22 | 140 | 1 | 1 | |
| | Rural | 80 | 66 | 7.71 (4.43–13.44) | **3.41 (1.58–7.39)** | **0.002**\* |
| Religion | Protestant | 62 | 112 | 1 | 1 | |
| | Islam | 23 | 44 | 0.94 (0.52–1.71) | 0.56 (0.21–1.51) | 0.251 |
| | Orthodox | 5 | 38 | 0.24 (0.09–0.64) | 0.69 (0.18–2.66) | 0.587 |
| | Others | 12 | 12 | 1.81 (0.77–4.26) | 2.40 (0.66–8.78) | 0.186 |
| Ever had physical IPV | Yes | 81 | 73 | 7.03 (4.02–12.28) | **2.35 (1.06–5.17)** | **0.034**\* |
| | No | 21 | 133 | 1 | 1 | |
| Ever had emotional IPV | Yes | 76 | 84 | 4.25 (2.51–7.18) | 2.12 (0.91–4.94) | 0.082 |
| | No | 26 | 122 | 1 | | |
| Parity | Primipara | 10 | 58 | 1 | 1 | |
| | Low multipara | 35 | 123 | 1.65 (0.76–3.56) | 2.81 (0.92–8.53) | 0.069 |
| | Grand multipara | 57 | 25 | 13.22 (5.83–30.01) | **5.36 (1.68–17.08)** | **0.005**\* |
| FP before recent pregnancy | Yes | 23 | 124 | 1 | 1 | |
| | No | 79 | 82 | 5.19 (3.02–8.93) | **5.82 (2.49–13.60)** | **< 0.001** |
| Experience of obstetric complications | Yes | 22 | 62 | 1 | 1 | |
| | No | 80 | 144 | 1.57 (0.89–2.74) | 1.01 (0.44–2.31) | 0.978 |
| Knowledge of danger signs of pregnancy | Poor | 73 | 80 | 4.98 (2.55–9.75) | 1.97 (0.71–5.52) | 0.195 |
| | Fair | 16 | 55 | 1.59 (0.71–3.58) | 1.28 (0.40–4.08) | 0.679 |
| | Good | 13 | 71 | 1 | 1 | |
| Time taken to a health facility | <30 minutes | 31 | 138 | 1 | 1 | |
| | ≥30 minutes | 71 | 68 | 4.65 (1.79–7.76) | **2.14 (1.02–4.51)** | **0.045**\* |
| Perceived quality of services | Good | 66 | 156 | 1 | 1 | |
| | Poor/don't know | 36 | 50 | 1.70 (1.02–2.85) | 2.23 (0.95–5.24) | 0.065 |
| Lack of transport due to COVID-19 | Yes | 32 | 28 | 2.91 (1.63–5.18) | 2.22 (0.90–5.49) | 0.085 |
| | No | 70 | 178 | 1 | | |
| Lack of face mask | Yes | 69 | 52 | 6.19 (3.68–10.42) | **2.69 (1.25–5.77)** | **0.011**\* |
| | No | 33 | 154 | 1 | 1 | |

Hosmer & Lemeshow goodness of fit: P-value = 0.913

Note: \*significant at p<0.05

AOR: Adjusted Odd Ratio

COR: Crude Odd Ratio

FP: Family Planning

IPV: Intimate Partner Violence

facilities, particularly long distances coupled with lack of transportation, poor road networks, and hills make use of these services less likely in rural areas. Rural women might be denied autonomy and access to media and information more than urban residents [16]. Moreover, different traditional and cultural beliefs that support TBAs and contradict the use of modern health services for childbirth are more pronounced among rural women than urban residents [17, 20, 22]. In most instances, women in rural areas have irregular income from agriculture or livestock rearing; as a result, they might lack readily available money to pay for indirect costs associated with transportation and food for her and accompanies during labour.

Women who needed more than or equal to 30 minutes to reach the nearest health facility were two times more likely to deliver at home than women who needed less than 30 minutes [AOR: 2.14; 95%CI: 1.02–4.51]. This result is consistent with several studies conducted in developing countries, including Nigeria, Ghana, and Ethiopia [17, 20, 24, 27, 30]. It is important to note that health services may be available within the community although this alone may not guarantee its use; therefore, healthcare facility delivery can be hampered by several factors associated with accessibility. Physical proximity of healthcare facilities and improved road network is the crucial enabling factor that enhances women's access to and use of health services, particularly in rural areas [20]. Long distances to health facility can be identified as one of the direct barriers that can hinder delivery services utilization due to unavailability and high costs of transport [42].

Among COVID-19 related factors, the women's non-use or lack of masks was significantly associated with homebirth. Women who did not use face masks during the COVID-19 pandemic were about 2.7 times more likely to have homebirth than their counterparts [AOR: 2.69; 95%CI: 1.25–5.77]. This result agreed with the study conducted in West Shoa Zone, Central Ethiopia [43]. This could be explained as women who lack a face mask have been disallowed from entering a health facility for maternal health services utilization like ANC and even institutional delivery because the use of face mask has been one of the compulsory measures to prevent transmission of COVID-19 infection from person to person. Hence, women who experience such challenges associated with a lack of money to buy a face mask during pregnancy might incline to deliver at home.

## Strengths and limitations of the study

The main strength of this study is, this study found COVID-19 related factors as an independent predictor of home birth, adding knowledge to the existing body of literature. This study has certain limitations; firstly, the selection bias of facility-based study was inevitable as those who attend public health facilities for any reason are different from those who do not. In addition, there can be a risk of under-reporting of IPV due to its sensitive nature, especially sexual violence by both cases and controls, possibly leading to non-differential misclassification bias. In addition, this research would have explored more about the determinants of home delivery if supplemented with a qualitative study. Due to its retrospective nature, the association between the predictors and outcome had no temporal relationship; thus causality was not established.

## Conclusions and recommendations

This study confirmed that the odds of home delivery were significantly higher among women in rural areas, grandmultiparity, ever experience of physical IPV, and women who lived more than 30 minutes walking distance from the nearest health facility. Furthermore, the COVID-19 pandemic significantly impacted women's use of maternal health services. The lack of personal protective equipment was a significant barrier to using institutional delivery.

The strategies that adapt to emerging public health threats, such as the COVID-19 pandemic, should be laid out to prevent their devastating effect on routine health services like maternal delivery service. Program planners should focus on the geographic accessibility of health facilities, and special attention should be given to women living in urban areas and grandmultiparity. We recommend further study using qualitative research targeting how IPV affects home delivery to devise a comprehensive intervention to reduce the experience of IPV and its negative impacts.

## Supporting information

**S1 File. English version questionnaire.**
(DOCX)

**S2 File. Sidaamu Afoo version questionnaire.**
(DOCX)

**S3 File. Raw SPSS data.**
(SAV)

## Acknowledgments

We are very thankful to Hawassa University for giving us the opportunity to conduct this study. We acknowledge Wondo Genet Health Office for their cooperation and for providing pertinent information for this study. We are also grateful to the supervisors, data collectors, and all mothers who participated in the study.

## Author Contributions

**Conceptualization:** Asaminew Geremu Gaga, Teshome Abuka Abebo, Yilkal Simachew.

**Data curation:** Asaminew Geremu Gaga, Teshome Abuka Abebo, Yilkal Simachew.

**Formal analysis:** Asaminew Geremu Gaga, Teshome Abuka Abebo, Yilkal Simachew.

**Funding acquisition:** Asaminew Geremu Gaga.

**Investigation:** Asaminew Geremu Gaga, Teshome Abuka Abebo, Yilkal Simachew.

**Methodology:** Asaminew Geremu Gaga, Teshome Abuka Abebo, Yilkal Simachew.

**Project administration:** Asaminew Geremu Gaga.

**Resources:** Asaminew Geremu Gaga.

**Software:** Asaminew Geremu Gaga, Teshome Abuka Abebo, Yilkal Simachew.

**Supervision:** Teshome Abuka Abebo, Yilkal Simachew.

**Validation:** Asaminew Geremu Gaga, Teshome Abuka Abebo, Yilkal Simachew.

**Visualization:** Asaminew Geremu Gaga, Teshome Abuka Abebo, Yilkal Simachew.

**Writing – original draft:** Asaminew Geremu Gaga, Yilkal Simachew.

**Writing – review & editing:** Asaminew Geremu Gaga, Teshome Abuka Abebo, Yilkal Simachew.

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
