## [Decision Letter · Decision Letter 0]

21 Nov 2022

PONE-D-22-17306Predictors of homebirth amidst COVID-19 pandemic among women attending health facilities in Wondo Genet, Sidama Region, Ethiopia: a case control studyPLOS ONE

Dear Dr. hunegnaw,

Thank you for submitting your manuscript to PLOS ONE. After careful consideration, we feel that it has merit but does not fully meet PLOS ONE’s publication criteria as it currently stands. Therefore, we invite you to submit a revised version of the manuscript that addresses the points raised during the review process.

The authors have made a nice effort to work on an important issue. The reviewers have a few comments that the authors are invited to address before we can make the final decision. In addition, please precisely edit the whole manuscript for English language errors, and provide definitions of the abbreviations in the first points of their uses.

We look forward to receiving your revised manuscript.

Kind regards,

Shabnam Iezadi, Ph.D.

Academic Editor

PLOS ONE

https://journals.plos.org/plosone/s/fileid=ba62/PLOSOne_formatting_sample_title_authors_affiliations.pdf.

3. In the ethics statement in the Methods, you have specified that verbal consent was obtained. Please provide additional details regarding how this consent was documented and witnessed, and state whether this was approved by the IRB

Reviewers' comments:

Reviewer's Responses to Questions

**Comments to the Author**

1. Is the manuscript technically sound, and do the data support the conclusions?

Reviewer #1: Partly

Reviewer #2: Yes

2. Has the statistical analysis been performed appropriately and rigorously? 

Reviewer #1: Yes

Reviewer #2: I Don't Know

3. Have the authors made all data underlying the findings in their manuscript fully available?

Reviewer #1: Yes

Reviewer #2: Yes

4. Is the manuscript presented in an intelligible fashion and written in standard English?

Reviewer #1: Yes

Reviewer #2: Yes

5. Review Comments to the Author

Reviewer #1: the theoretical framework of matched sampling and that of logistic regression may be expanded a little more. also, there are typos which need to be addressed, for example, lines 57-58 need to be re-framed as "The devastating effect of coronavirus disease 2019 pandemic on maternity services should be prevented.

Reviewer #2: Dear author, thank you for submitting this work as home birth is common problem in low-income countries, and lead to adverse health outcomes of both the mothers and their babies. Overall, you did great but there are points that need revision. Here are my comments and questions.

General comment on method: - case control study design is preferred study design in case of rare disease (outcome of interest) while home delivery is very common scenario particularly in low-income countries like Ethiopia. So why the author prefers this study designs over others e.g., correctional study design. I suggest the author at least to discuss it briefly.

1. methods section: what about mothers who might gave birth at private health facilities?? Did you included or excluded from the study? Please explain it.

2. methods section: in your sampling procedure, you stated that you selected three health facilities out of five health facilities using simple random sampling technique, and then take sample of mothers from these HF. What was the sampling method used to select study participants from the three HFs? You didn’t explain anything about muti stage sampling kindly give explanation about it.

3. methods section: As you know there are three control selection methods in Case-control studies’ namely: Survivor sampling, Case-base or case-cohort sampling and Risk set sampling. kindly give the control selection method you used.

4. discussion section: your conclusion and recommendation should be based on your finding. For example, in the conclusion part you stated that “long distances coupled with poor road systems play a significant role in most 555 rural areas…” But you didn.t used road system variable in your analysis. Additionally, you conclude that “Gender inequalities denying women’s independent decision making, education, and freedom of movement restrict use of maternal health services including facility delivery”. Please conclude based on what you found in your statistical analysis.

5. discussion section: kindly give the strength of this study

6. PLOS authors have the option to publish the peer review history of their article (what does this mean?). If published, this will include your full peer review and any attached files.

Reviewer #1: **Yes: **Emmanuel T. Adewuyi

Reviewer #2: **Yes: **Bayley Adane Takele

---

## [Author Response · Author response to Decision Letter 0]

3 Jan 2023

REPLY TO ACADEMIC EDITOR’S COMMENTS

Response: Checked and corrected; we have now reformatted the manuscript (font size, font style, line spacing, figure caption, table caption, reference citation and file naming) according to the guidelines and style requirements of PLOSE ONE. 

Response: Our study subjects were mothers who gave birth in the last 12months prior to the study. The nine (2.9%) mothers less than 18 years old who were participated in this study were already married and do not live with the parents. Verbal consent to participate in the study was obtained from these mothers (3 cases and 6 controls) and approved by the IRB. The IRB is well aware that the respondents in this study were married and already left their parents. If a woman marries before the age of 18 years old, she leaves her parents and lives her life independently of her parents. The IRB is aware that this is a reality in Ethiopia and thus, it approved the consent without requiring additional parental consent. Based on the given comments, now we have stated this on method section (Revised manuscript, line #298-301).

3. In the ethics statement in the Methods, you have specified that verbal consent was obtained. Please provide additional details regarding how this consent was documented and witnessed, and state whether this was approved by the IRB

Response: Based on the suggestions, we clarify how the verbal consent was documented and witnessed, and statement about the IRB approval “The data collector read the information sheet and consent to the participants slowly and loudly. Then they were asked if they had any questions. After the mothers confirmed that everything is clear, they were asked one last time if they wanted to participate in the study. The data collectors circled either 'yes' or 'no' depending on which option was chosen, and the interview was only conducted if the data collector was instructed to circle the response 'yes'”. We have included the above detail under method section in “Ethical consideration” (Revised manuscript, line #292-297).

Response: We've updated the reference list to make it complete and accurate. We updated the references to conform to the International Committee of Medical Journal Editors' reference format (ICMJE). The following is a list of updated references to ensure that it is complete and accurate, as required by the journal.

Reference number 2, 4, 5, 7, 8, 10, 12, 14, 15, 17-27, and 31-46: According to ICMJE citation style, it is an online paper with a DOI. As a result, we have included the entire DOI next to the publication date in the revised manuscript. 

Reference number 2, 4, 7, 8, 15, 21, 23, 25, 26, 31, 32, 35, 37-39, 41-43, and 45: In the previous format the journal titles were presented in its full form, however according to the PLOS ONE guide journal titles should be abbreviated according to NLM Catalog of NCBI Journals. Accordingly we have revised the journals title in the revised manuscript. 

Reference number 3, 6, 9 and 30: The source is a website, but the accessed date and availability were missing in the previous format. We have made the necessary changes.

Reference number 1, 2, 4, 5, 7, 8, 10-29, 31-42, and 44-46: In the previous format the publication date was presented incorrectly. Now we have revised it by including the publication date in the right place next to abbreviated journal title.

Reference number 1, 2, 4, 7, 8, 18, 20, 24, 32, 39 and 46: In the previous format the authors list were not presented according to the reference style used by PLOS ONE journal. Now we have revised it by listing the authors surname first followed by their first and middle initials, in the revised manuscript. 

REPLY TO REVIEWERS’ COMMENTS

Comments from reviewer # 1

Reviewer #1: the theoretical framework of matched sampling and that of logistic regression may be expanded a little more. also, there are typos which need to be addressed, for example, lines 57-58 need to be re-framed as "The devastating effect of coronavirus disease 2019 pandemic on maternity services should be prevented.

Response: We thank the reviewer for useful comment and suggestion. We agree with the reviewer that the theoretical framework of matched sampling and logistic regression is expanded a little more in our manuscript. Now, we have revised it to make it short and precise (Revised manuscript, line #252-258, line#265-267, line#270-271). In addition, we have corrected the typos error (Revised manuscript, line #57).

Comments from reviewer # 2

Reviewer #2: Dear author, thank you for submitting this work as home birth is common problem in low-income countries, and lead to adverse health outcomes of both the mothers and their babies. Overall, you did great but there are points that need revision. Here are my comments and questions.

Response: Thank you very much for your nice wording. 

General comment on method: - case control study design is preferred study design in case of rare disease (outcome of interest) while home delivery is very common scenario particularly in low-income countries like Ethiopia. So why the author prefers this study designs over others e.g., correctional study design. I suggest the author at least to discuss it briefly.

Response: We thank the reviewer for raising critical question. As the reviewer mentioned case control is preferred study design for rare disease, and home delivery is very common practice particularly in our country context. However, as we have mentioned in the background section, the prevalence of home delivery is well studied and known in our setting, even one can get about the exact figure of home delivery from the district health office. However, there are limited studies which investigate the factors that affect home birth. Because of this reasons our study objective was to identify multiple factors that affect home birth and Case-control study is an efficient research method for investigating risk factors of home birth. We believe that case control is appropriate for the identified research problem and the best option for answering our research question.

1. methods section: what about mothers who might gave birth at private health facilities?? Did you included or excluded from the study? Please explain it.

Response: Thank you for raising an important point here. Before conducting our study during proposal phase, we have considered to include mothers who gave birth at private health facilities to avoid selection bias. However, despite there are some private clinics (medium and primary) at wondo Genet, none of them provide delivery service. That is why we didn’t consider it. 

2. methods section: in your sampling procedure, you stated that you selected three health facilities out of five health facilities using simple random sampling technique, and then take sample of mothers from these HF. What was the sampling method used to select study participants from the three HFs? You didn’t explain anything about muti stage sampling kindly give explanation about it.

Response: We apologize for lack of clarity in our first manuscript regarding the sampling method we have used after selecting three health facilities out of five. First we have selected three health facilities out of five using simple random sampling technique. Then, the sample size was proportionally allocated based on the client flow for immunization and post natal services at each of the three health facilities in the year proceeding the study period. Mothers who give birth during last 12 months and visited the selected health facilities were classified as case and control based on where they delivered. Because of difficulty for preparing an exact sample frame of case and control based on health facility reports of immunization and post natal care (PNC) service, and the estimated number of cases and controls for data collection period (one month) were not large enough, we used the consecutive sampling method. Based on the give comment, we have included a sentence that describes the sampling method we used to select study participants from three health facilities (Revised manuscript, line# 199).

3. methods section: As you know there are three control selection methods in Case-control studies’ namely: Survivor sampling, Case-base or case-cohort sampling and Risk set sampling. kindly give the control selection method you used.

Response: we would like to thank the reviewer for the detail comment. As correctly mentioned by the reviewer there are three control selection methods. Among these methods we have used survivor sampling or cumulative density sampling because controls are selected from those who remain free of the outcome variable (home delivery) or from those who give birth at health facility during 12 months before the data collection period. It is not risk set sampling or incident sampling technique because in this sampling method the controls are sampled at the time of the occurrence of a case, and we usually use this method when the cases are incident cases. But our study used prevalent case since we considered mothers who had given birth in the last 12 months before the data collection period. In addition, it is not also case based or case cohort sampling because this type of sampling only works with a previously defined cohort where controls are selected from the population at risk at the beginning of the follow-up period in the cohort study. 

4. discussion section: your conclusion and recommendation should be based on your finding. For example, in the conclusion part you stated that “long distances coupled with poor road systems play a significant role in most 555 rural areas…” But you didn.t used road system variable in your analysis. Additionally, you conclude that “Gender inequalities denying women’s independent decision making, education, and freedom of movement restrict use of maternal health services including facility delivery”. Please conclude based on what you found in your statistical analysis.

Response: We thank the Reviewer for the critical review and the useful suggestions. Comment is accepted and correction is made (Revised manuscript, line #574-587).

5. discussion section: kindly give the strength of this study

Response: Thank you for your suggestion; accordingly we have incorporated your suggestion (Revised manuscript, line #558-561).

---

## [Editor Report · Decision Letter 1]

31 Jan 2023

PONE-D-22-17306R1Predictors of homebirth amidst COVID-19 pandemic among women attending health facilities in Wondo Genet, Sidama Region, Ethiopia: a case control studyPLOS ONE

Dear Dr. hunegnaw,

Thank you for submitting your manuscript to PLOS ONE. After careful consideration, we feel that it has merit but does not fully meet PLOS ONE’s publication criteria as it currently stands. Therefore, we invite you to submit a revised version of the manuscript that addresses the points raised during the review process.

ACADEMIC EDITOR:Dear authors,Thank you for making the necessary revisions. There are just a few points that need to be addressed before we can make the final decision.1-There are still some typo errors, line 286 (76.6% %) for example. Please, kindly review the whole manuscript and edit all typos. 2- In line 184 in sample size and sampling procedures section, reference number 61 does not exist in the reference list. Please check this out and kindly review the references for any mismatch. 3- Please, add a statement to the text to indicate that "despite there are some private clinics (medium and primary) at wondo Genet, none of them provide delivery service". Although you have explained this in your response letter, there is no change in the body of the manuscript to show your explanation. 4- Please, add a statement to the text (method section) about the control selection sampling method (survivor sampling or cumulative density sampling). Although you have explained this in your response letter, there is no change in the body of the manuscript to show your explanation.

We look forward to receiving your revised manuscript.

Kind regards,

Shabnam Iezadi, Ph.D.

Academic Editor

PLOS ONE
---

## [Author Response · Author response to Decision Letter 1]

7 Mar 2023

REPLY TO THE ACADEMIC EDITOR’S COMMENTS

Comment 1: Dear authors, Thank you for making the necessary revisions. There are just a few points that need to be addressed before we can make the final decision. 1-There are still some typo errors, line 286 (76.6% %) for example. Please, kindly review the whole manuscript and edit all typos.

Response: Thank you for advising us to review the whole manuscript and edit the typos errors. Now, we have revised and edited the typos errors throughout the manuscript. The changes that have been made are found in the revised manuscript with track changes. 

Comment 2: In line 184 in sample size and sampling procedures section, reference number 61 does not exist in the reference list. Please check this out and kindly review the references for any mismatch.

Response: Thank you for pointing this out. To calculate the sample size, we took those parameters from the study conducted in Northern Ethiopia with reference number 44 (Tsegay R, Aregay A, Kidanu K, Alemayehu M, Yohannes G. Determinant factors of home delivery among women in Northern Ethiopia: a case control study. BMC Public Health. 2017 Apr 4;17(1):289. doi: 10.1186/s12889-017-4159-1). Now, we have revised it accordingly (Revised manuscript, line #187). 

Comment 3: Please, add a statement to the text to indicate that "despite there are some private clinics (medium and primary) at wondo Genet, none of them provide delivery service". Although you have explained this in your response letter, there is no change in the body of the manuscript to show your explanation.

Response: We have added the above statement in the revised manuscript under the method and materials section in the study area (Revised manuscript, line #164-165). 

Comment 4: Please, add a statement to the text (method section) about the control selection sampling method (survivor sampling or cumulative density sampling). Although you have explained this in your response letter, there is no change in the body of the manuscript to show your explanation.

Response: We have included a statement about the control selection sampling method under the method section in the sample size and sampling procedure subsection (Revised manuscript, line #198-200).

---

## [Editor Report · Decision Letter 2]

13 Mar 2023

Predictors of homebirth amidst COVID-19 pandemic among women attending health facilities in Wondo Genet, Sidama Region, Ethiopia: a case control study

PONE-D-22-17306R2

Dear Dr. hunegnaw,

We’re pleased to inform you that your manuscript has been judged scientifically suitable for publication and will be formally accepted for publication once it meets all outstanding technical requirements.

Kind regards,

Shabnam Iezadi, Ph.D.

Academic Editor

PLOS ONE

Additional Editor Comments (optional):

Please kindly provide definitions for COR and AOR in the legend of the table 4 and remove AOR and CIs where discussing your results in the discussion section.
---

## [Editor Report · Acceptance letter]

24 Apr 2023

PONE-D-22-17306R2 

Predictors of homebirth amidst COVID-19 pandemic among women attending health facilities in Wondo Genet, Sidama Region, Ethiopia: a case control study 

Dear Dr. Simachew:

I'm pleased to inform you that your manuscript has been deemed suitable for publication in PLOS ONE. Congratulations! Your manuscript is now with our production department. 

Kind regards, 

on behalf of

Dr. Shabnam Iezadi 

Academic Editor

PLOS ONE